# Transcriptome-Based WGCNA Analysis Reveals the Mechanism of Drought Resistance Differences in Sweetpotato (*Ipomoea batatas* (L.) Lam.)

**DOI:** 10.3390/ijms241814398

**Published:** 2023-09-21

**Authors:** Jikai Zong, Peitao Chen, Qingqing Luo, Jilong Gao, Ruihua Qin, Chunli Wu, Qina Lv, Tengfei Zhao, Yufan Fu

**Affiliations:** Engineering and Technology Research Center for Sweetpotato of Chongqing, School of Life Science, Southwest University, Chongqing 400715, China; zongjikai@126.com (J.Z.); capital0529@163.com (P.C.); luoqvqing@163.com (Q.L.); 13598024606@163.com (J.G.); qrh15199576713@163.com (R.Q.); wcl98248654@163.com (C.W.); 17823059244@163.com (Q.L.); tengfeizhao@swu.edu.cn (T.Z.)

**Keywords:** sweetpotato, drought resistance, RNA-Seq, WGCNA analysis, physiological response

## Abstract

Sweetpotato (*Ipomoea batatas* (L.) Lam.) is a globally significant storage root crop, but it is highly susceptible to yield reduction under severe drought conditions. Therefore, understanding the mechanism of sweetpotato resistance to drought stress is helpful for the creation of outstanding germplasm and the selection of varieties with strong drought resistance. In this study, we conducted a comprehensive analysis of the phenotypic and physiological traits of 17 sweetpotato breeding lines and 10 varieties under drought stress through a 48 h treatment in a Hoagland culture medium containing 20% PEG6000. The results showed that the relative water content (RWC) and vine-tip fresh-weight reduction (VTFWR) in XS161819 were 1.17 and 1.14 times higher than those for the recognized drought-resistant variety Chaoshu 1. We conducted RNA-seq analysis and weighted gene co-expression network analysis (WGCNA) on two genotypes, XS161819 and 18-12-3, which exhibited significant differences in drought resistance. The transcriptome analysis revealed that the hormone signaling pathway may play a crucial role in determining the drought resistance in sweetpotato. By applying WGCNA, we identified twenty-two differential expression modules, and the midnight blue module showed a strong positive correlation with drought resistance characteristics. Moreover, twenty candidate Hub genes were identified, including *g47370* (*AFP2*), *g14296* (*CDKF*), and *g60091* (*SPBC2A9*), which are potentially involved in the regulation of drought resistance in sweetpotato. These findings provide important insights into the molecular mechanisms underlying drought resistance in sweetpotato and offer valuable genetic resources for the development of drought-resistant sweetpotato varieties in the future.

## 1. Introduction

Sweetpotato (*Ipomoea batatas* (L.) Lam.) is widely recognized as a crucial crop that contributes significantly to global food production and is a highly competitive energy crop due to its impressive yield, short growth cycle, exceptional ability to withstand famine, and its rich nutritional content [1]. In addition, sweetpotato storage roots and leaves are recognized for their numerous health benefits, such as their antitumor, antioxidant, and free-radical scavenging effects. As a result, the consumption and utilization of sweetpotato in both food consumption and food processing are steadily growing [2]. Sweetpotatoes are commonly grown in hilly or marginal areas where irrigation resources are limited or where dry weather conditions prevail. As a result, they are often exposed to drought stress during their growth stage. An extreme and prolonged drought could lead to significant production losses, with yield reductions exceeding 30% in various regions worldwide [3].

Therefore, researching the sweetpotato’s molecular mechanisms of drought tolerance and further breeding drought-resistant varieties are of great significance for maintaining sweetpotato yield and quality. Excellent drought-tolerant genetic resources play a crucial role in breeding drought-tolerant varieties. For example, Jia et al. achieved better drought tolerance in sweetpotato by hybridizing it with closely related species that possess drought-resistant characteristics [4]. Meanwhile, exploring endogenous drought-responsive genes in plants using molecular biology techniques can enhance the efficiency and accuracy of creating superior resources and breeding varieties for drought resistance, exhibiting important potential for practical implementation. For example, in *Artemisia annua*, Shu et al. identified an ABA-responsive transcription factor, bZIP, which plays a crucial role in enhancing drought tolerance. Overexpression of the bZIP gene significantly improved the drought resistance in *A. annua* [5]. Therefore, it is of great research significance to screen drought-resistant germplasm resources in sweetpotato and explore the endogenous genes involved in their drought response. This will help to reveal the molecular mechanisms underlying drought stress in sweetpotato and facilitate the breeding of new drought-tolerant sweetpotato varieties.

Artificially simulating drought conditions is a widely used approach to studying plant drought tolerance [6,7]. Polyethylene glycol (PEG) is a non-reactive and excellent osmotic agent that is commonly employed for regulating permeability. The application of a PEG solution treatment provides a reliable method for simulating soil drought conditions and investigating the mechanisms of drought resistance in various crops, including tomato, wheat, and rice. The results consistently demonstrate the effectiveness of the PEG treatment in studying plant drought tolerance [8,9,10]. The drought stress experienced by plants can trigger a series of physiological changes in the water content, chlorophyll levels, osmotic protectants, enzymatic or non-enzymatic antioxidants, membrane peroxidation, and other primary and secondary metabolism in plants. These changes are closely linked to the plant’s ability to withstand drought stress [11,12,13], and some of these substances’ changes could serve as indicators of the drought-tolerance capability of a plant. Because plants can promote root growth and maintain cell expansion in order to find deep water sources under drought conditions [14,15], and that by closing stomata and reducing transpiration, blade water loss can be reduced, we chose the following three indicators: the relative water content (RWC%) in the blade, the vine-tip fresh-weight reduction (VTFWR%), and the number of rootings (R) of the vine tip. The contents of malondialdehyde (MDA), soluble protein (SP), superoxide anion (O^2−^), and chlorophyll (CHL) are the key indexes for evaluating the degree of drought tolerance [16]. Because drought stress conditions lead to an increased production of reactive oxygen species, disrupting the cellular reduction–oxidation regulation [17,18], plants under drought stress will produce many osmoregulatory substances, such as soluble protein, soluble sugar, proline, and other substances, and these substances play a role in maintaining cell osmotic balance, stabilizing the structure and function of biological macromolecules, and scavenging free radicals. Photosynthetic activity decreases due to closing stomata, membrane damage, and related enzyme functional changes, reducing carbohydrate accumulation [19]. 

RNA sequencing (RNA-seq) is a highly sensitive and accurate genome technology for profiling mRNA expression. It is commonly employed to uncover variations in the plant transcriptome, particularly the physiological metabolism, stress response, and other related aspects, which are then used to identify the key genes involved in these processes. For example, Zhang et al. performed a transcriptome analysis that revealed that the sugar–starch conversion step catalyzed by sucrose synthase and UDP-glucose pyrophosphorylase in the sucrose metabolism of sweetpotato may play a crucial role in starch accumulation [20]. Chen et al. employed a comparative transcriptome analysis, which indicated that the phenylpropanoid pathway is likely a significant pathway contributing to the differences of phenolic contents in blades among sweetpotato genotypes [21]. RNA-seq and other biotechnological tools provide a convenient way to identify gene expression changes in response to environmental stimuli and facilitate the screening of candidate genes. Validating these candidate genes through functional genomics approaches will aid excellent germplasm creation and stress-tolerant variety breeding [22]. Indeed, applying weighted gene co-expression network analysis (WGCNA) can help to identify highly correlated gene modules for the growth, development, and physiological processes of plants, along with their Hub genes. For instance, EI-Sharkawy et al. conducted a transcriptome analysis of yellow-fruited mutants and identified a gene network module highly correlated with epigenetic regulation of anthocyanins, and 22 related genes were reported [23]. Cai et al. found that a transcription factor named “Tai6.25300” is closely associated with storage root enlargement by a comparative transcriptome analysis of two sweetpotato genotypes [24].

At present, there are few studies on the drought-resistance mechanism and germplasm resource screening of sweetpotato using a comparative transcriptome analysis. In this study, two genotypes were screened out with different levels of drought tolerance from the comprehensive analysis of the morphological and physiological indexes of 27 sweetpotato genotypes under PEG-solution-simulated drought conditions. The differences in the resistance mechanisms to drought in two sweetpotato genotypes were further explored using an RNA-Seq analysis. Using WGCNA technology to associate physiological indicators such as the RWC, VTFWR, and MDA with gene expression patterns, we extracted the Hub genes directly related to drought defense and constructed a co-expression network. This study not only screened and confirmed some drought-tolerant sweetpotato germplasms but also provides a reference for understanding the molecular mechanism of sweetpotato under drought stress. This evidence also lays the foundation for the breeding of sweetpotato drought-tolerant varieties.

## 2. Results 

### 2.1. Selection of Drought-Resistant Sweetpotato Genotypes

In order to select genotypes with strong drought resistance from 27 sweetpotato genotypes, we measured the relative water content (RWC%) in the blade, the vine-tip fresh-weight reduction (VTFWR%), and the number of rootings (R) of the vine tip. The results of the RWC, VTFWR around the blades, and R from 27 sweetpotato genotypes were visualized using a heatmap (Figure 1a). The results showed that each index had a significant difference among the 27 genotypes (Appendix A), and the three indexes had an obvious positive relationship. A correlation analysis among the RWC, VTFWR, and R showed a significant positive correlation among them (Figure 1b). There was no difference between the five genotype types in their RWC, VTFWR, and R averages. Among the 27 sweetpotato genotypes, the RWC and VTFWR of S01 (XS161819) ranked first, reaching 0.70 and 0.87, respectively, and the R ranked second, reaching 25.3. The R value of S27 (Chaoshu-1) ranked first, reaching 28.33. The other two indexes of S27, which is a recognized drought-resistant variety [25], were also at a high level. The RWC, VTFWR, and R of S26 (18-12-3) all ranked last at 0.41, 0.68, and 6.67, respectively. The RWC, VTFWR, and R of S01 were 1.71 times, 1.28 times, and 3.79 times that of S26. A PCA analysis of the 27 genotypes (Figure 1c) showed that S01 and S26 were outliers among all genotypes. In summary, compared with Chaoshu-1 with drought resistance, S01, S02, S05, and S08 had a highly comprehensive drought tolerance, which belonged to genotypes type 1, 2, 3, and 4, respectively.

### 2.2. Physiological Responses of Sweetpotato Genotypes S01 and S26 to Drought Stress

In order to further verify the drought tolerance level of the sweetpotato S01 and S26 genotypes, the physiological indexes of S01 and S26 were measured, and the phenotypic changes were observed before and after the 20% PEG6000 treatment. After a 48 h 20% PEG6000 treatment, compared with the control group, the contents of malondialdehyde (MDA), soluble protein (SP), and superoxide anion (O^2−^) in the blades of S01 and S26 were all significantly increased (Figure 2a); however, the absolute increases in the MDA, SP, and O^2−^ contents in S26 were 7.21 times, 1.46 times, and 2.62 times more than those in S01, respectively, and the increased percentages of the MDA, SP, and O^2−^ contents in S26 were 6.72 times, 2.26 times, and 4.16 times than those in S01, respectively. The absolute reduction in CHL A and CHL B in S26 was 3.97 times, and 3.76 times those in S01, respectively, and the reduction percentage of CHL A and CHL B in S26 was 2.77 times, and 2.42 times those in S01. The blade phenotypic observation of S26 showed yellowing, shriveling, and water loss caused by the drought treatment, while S01 had little change (Figure 2b). From the overall phenotype of the plant, part of the leaves of S26 were dry and the roots did not grow, while the leaves of S01 had only a slight water loss and the roots were still growing (Appendix A). In summary, S01 was a drought-resistant genotype, and S26 was a drought-sensitive genotype. Therefore, a subsequent RNA-seq analysis and a WGCNA analysis were performed on S01 and S26.

### 2.3. Transcriptome Sequencing Analysis 

The Illumina Nova Seq 6000 sequencing platform was used to sequence 12 samples and obtain 82.14 Gb of raw data. In order to ensure the quality and reliability of the data analysis, 78.46 Gb of clean data for subsequent analysis were obtained by filtering the original data by checking the sequencing error rates. In addition, Q20 ≥ 96.4% and Q30 ≥ 90.63% were identified in all samples (Appendix A); using HISAT2 software (2.0.5), clean reads and reference genomes were quickly and accurately compared, and an average of 81.68% were matched to the reference genome. The analysis of transcriptome data using Pearson correlation coefficient showed that the three biological replicates had good consistency and met the requirements for the subsequent analysis (Appendix A).

### 2.4. Differentially Expressed Genes (DEGs) Analysis

To find differentially expressed genes with KEGG critical pathways, firstly, the expression data of S01 and S26 were statistically analyzed. After multiple hypothesis testing and correction, 12,839 differentially expressed genes were screened. There were many differentially expressed genes between different combinations (Figure 3). The S01-P (S01 experimental group) and S26-P (S26 experimental group) had the most DEGs, with 4594 up-regulated and 3509 down-regulated genes. The GO enrichment results showed that differential genes were enriched and annotated into “biological processes”, “cell recognition”, and “molecular functions”. The GO enrichment analysis results of the S01 in the experimental group and the control group were like those of S26, mainly for “biological processes” and “cellular processes”. The group with the highest proportion of “biological processes” had “photosynthesis”. The group with the highest percentage of “cellular processes” was called thylakoid. The GO enrichment results of S01 and S26 were compared in the same treatment group, mainly for “molecular function”. The group with the highest proportion of “molecular function” was divided into “carbon-oxygen lyase activity”, followed by “terpene synthase activity” and “acting on phosphates” (Appendix A).

The KEGG metabolic pathway was analyzed, and the results are shown in Appendix A. The differences in the KEGG enrichment between S01_CK and S01_P, and S26_CK and S26_P were analyzed. The main pathways were “Plant hormone signal transduction”, “Carbon metabolism”, “Biosynthesis of amino acids”, and “Starch and sucrose metabolism”. The differences in the KEGG enrichment between S01_CK and S26_CK, and S01_P and S26_P were analyzed. The main pathways were “Plant hormone signal transduction”, “Amino acid biosynthesis”, and “Phenylpropane biosynthesis”. The “Plant hormone signal transduction plays” an important role in the drought resistance of sweetpotato, especially abscisic acid (ABA) and the auxin signal transduction pathway.

### 2.5. Key Gene Modules for Drought Tolerance Screened by WGCNA

In order to further explore the relationship between traits and DEGs, genes with an FPKM (Fragments Per Kilobase of transcript sequence per Millions of base pairs sequenced) expression less than one were filtered, and the remaining genes were used for WGCNA analysis. The tree of gene clusters was successfully constructed, and 22 stable expression modules were obtained (Figure 4a). The number of genes in each module varied greatly, ranging from 43 to 1944 (Figure 4b). In order to find the key modules related to drought tolerance, the modular–feature relationship was analyzed (Figure 4c). The midnight blue modules were positively correlated with MDA, SP, O^2−^, and the 20% PEG6000 treatment, and the correlation coefficients were 0.87, 0.71, 0.85, and 0.92, respectively. The midnight blue modules were negatively correlated with the RWC, vtfwr, Chl a, Chl b, and Hoagland medium, and the correlation coefficients were −0.81, −0.82, −0.82, −0.84, and −0.92, respectively.

### 2.6. Hub Genes for Drought Tolerance Screened by WGCNA

To identify Hub genes associated with drought tolerance in the midnight blue modules, gene network analysis was performed using Cytoscape software (3.9.1) (top 1000 edges). After removing the genes with a betweenness centrality of 0, the top 20 genes with the greatest betweenness centrality with other genes were selected as “Hub genes” and shown as red nodes (Figure 5). Appendix A provides more information on the “Hub genes” of the midnight blue modules. Based on previous reports, multiple homologs of these genes were found to be involved in plant biotic stress and stress-resistance processes [26,27,28,29,30,31].

*G47370* (*AFP2*), *g14296* (*CDKF*), *g60091* (*SPBC2A9*), *Novel. 244*, and *g38080*(*PVA22*) are not only key genes in drought response, but also have higher expression levels in S01_CK than S26_CK. Therefore, they may be the key genes that lead to the differences in drought resistance between S01 and S26.

### 2.7. qRTPCR Validation of RNA-Seq Data

To verify the accuracy of the results of the RNA-seq, nine genes of DEGs were randomly selected for the qRT-PCR analysis (Figure 6). The qRT-PCR results showed that the expression patterns of these nine DEGs were similar to their results in the RNA-seq (Appendix A).

## 3. Discussion

Sweetpotato is one of the most important cash crops and plays a role in relieving the problem of global food security. With the deterioration of the ecological environment and global climate anomalies, drought has become a common meteorological disaster. Although sweetpotato has a certain degree of drought resistance, a severe drought will lead to a serious yield reduction and even no harvest [32]. Drought tolerance varies greatly among different sweetpotato genotypes [33]; therefore, one of the most effective coping strategies for sweetpotato resistance to drought is through the breeding of drought-tolerant varieties based on the genetic ability of drought tolerance and excellent germplasm creation. Sweetpotato germplasm screening and variety breeding for resistance to drought have always been the most important tasks for our research group.

Our 17 sweetpotato breeding lines and 10 sweetpotato varieties released by other agricultural academies or institutes, including S27 (Chaochu-1) with drought resistance and S15 (Xushu-22) with drought sensitivity [34], were used for the drought-resistance appraisal in this study (Appendix A) using the common method of a 20% PEG6000 drought solution simulation drought. Through the RWC, VTFWR, and the R of the vine tip, some breeding lines or germplasms with strong drought resistance superior to S27 (Chaochu-1) were identified as drought-tolerant, including S01 (XS161819) and S04 (Mianshu-6) used for storage-root starch processing, S02 (XN1729-11) and S07 (Jinshu-3) used for storage-root eating consumption, S05 (21-F-3) and S10 (21-P-29) used for vine-tip eating consumption, and S08 (G20-9) with a purple-fleshed storage root. However, S15 (Xushu-22), S26 (18-12-3), S25 (21-P-26), etc., were drought-sensitive genotypes. The results of the analysis of the content changes in the MDA, SP, O^2−^, and CHL in S01 and S26 after the 20% PEG solution showed that there was a slower increase in the MDA, SP, and O^2−^, and a slower the CHL content decrease in S01 than those in S26; namely, S01 better combatted drought than S26. This is similar to the relevant research on wheat and Chinese hickory [31,32,33,34]. In the control group, the soluble protein content of S01 was much higher than that of S26, which may be one of the important reasons for the difference in the drought resistance between the two genotypes.

The results of the KEGG enrichment analysis of S26_P and S26_CK were similar to those of S01_P and S01_CK, and the differentially expressed genes of the abscisic acid and salicylic acid signaling pathways were significantly up regulated. The results are similar to those of Hsu et al. and Kang et al. Hsu et al. showed that the drought tolerance of plants could be improved by enhancing abscisic acid signaling [35]. Kang et al. found that a series of proteins such as glutathione *S*-transferase and dehydroascorbate reductase were differentially expressed in physiological and metabolic pathways related to drought stress in wheat seedlings that were treated with exogenous acid. These proteins confer a better growth and drought-tolerant phenotype on wheat [36]. We also found that “Carbon metabolism”, “Biosynthesis of amino acids”, and “Starch and sucrose metabolism” had a large number of differentially expressed genes. These pathways may play a positive role in plant osmotic regulation under drought conditions. The results of the KEGG enrichment analysis of S01_P and S26_P showed that there were many differentially expressed genes in the auxin and abscisic acid pathways (Appendix A). For example, the Aux/IAA-enhanced gene was significantly increased in S01. Aux/IAA proteins are a large family of auxin co-receptors and transcriptional inhibitors that play a central role in auxin signaling. This was similar to the research on sorghum, Arabidopsis, and rice, where Aux/IAA genes can enhance their drought resistance [37,38,39]. The expression of the PP2C (2C protein phosphatase) gene is significantly downregulated in S01, and PP2C is an important component of abscisic acid signaling. PP2C inhibits the activity of SnRK2 (Sucrose Nonfermenting 1-Related Protein Kinase 2) via dephosphorylation, thereby preventing abscisic acid signaling to reduce drought resistance [40]. Therefore, the hormone signaling pathway may be the key pathway leading to the difference in the drought resistance of sweetpotato.

In this study, 22 gene co-expression modules were constructed using a WGCNA analysis, among which the midnight blue modules showed a high positive correlation with drought resistance physiological characteristics of sweetpotato. We carried out a gene network analysis and selected the top 20 genes with the greatest betweenness centrality with other genes called “Hub genes”. Some of the Hub genes have been validated in other species and identified as related to abiotic stress. For example, the *CDKF* can enhance drought resistance in *Arabidopsis* by increasing the concentration of antioxidant enzymes and cell membrane stability under drought stress [26]. *CIPK3* is a calcium sensor-associated protein kinase, which can regulate ABA and cold signal transduction in *Arabidopsis* [28]. Based on comprehensive GWAS and transcriptomics analysis, *SYT3* and the other four most promising salt responsive genes were screened [41]. *ABI5* is a key regulator of ABA signaling and stress response, and *ABF2* can interact with *ABI5* to enhance plant resistance to abiotic stress [31]. Indirect explanations suggest that *g14296* (*CDKF*), *g50956* (*CIPK3*), *g57974* (*SYT3*), and *g47370* (*AFP2*) also have similar functions in sweetpotato. Other Hub genes not reported in other species, such as g9020, g49825, and g50524, may play an important role in improving the drought resistance of sweetpotato. Further experiments are needed to verify how these Hub genes functioned in sweetpotato. Nine genes were selected from the differentially expressed genes for qRT-PCR verification, and the results were consistent with the transcriptome expression level.

Focusing on the drought-resistant breeding of sweetpotato varieties, some drought-resistant breeding lines or excellent germplasms were selected and determined at the physiological and transcriptional levels in the present study, and potential drought-resistant-related genes were identified.

## 4. Materials and Methods

### 4.1. Plant Materials 

Twenty-seven sweetpotato genotypes (Appendix A) were provided by Chongqing Engineering Research Center for Sweetpotato, Southwest University, Chongqing, China. After 90 d of sweetpotato transplanting, 20 cm of fresh, disease-free, and healthy vine tips were hydroponically cultivated using 1/2 Hoagland medium and acclimated for 24 h in an artificial climate chamber. Then, some vine tips (experimental group) were transferred to 1/2 Hoagland culture medium containing 20% PEG6000 for 48 h, and the vine tips of the control group were left hydroponically in 1/2 Hoagland medium for 48 h. Each group contained 3 biological replicates. The third mature blades from each stem top per vine tip were sampled, immediately put into liquid nitrogen for rapid cooling, and stored at −80 °C for further analysis.

### 4.2. Determination of Water Loss and Counting Rooting Number after PEG Treatment

The relative water content (RWC%) in the blade was determined according to Sairam et al. [42]; RWC% = [(M1 − M3)/(M2 − M3)] × 100. In this formula, M1 represents the initial weight of the blade before treatment, M2 represents the weight of the blade after removing the surface water by immersing it in ultra-pure water for 12 h at room temperature, and M3 represents the constant weight obtained after drying the blade at 60 °C following an inactivated treatment at 115 °C for 5 min. The vine-tip fresh-weight reduction (VTFWR) is given as VTFWR = M5/M4, where M4 and M5 represent the weight of the vine tip before and after the 20% PEG6000 treatment, respectively. Meanwhile, the number of rootings (R) of the vine tip in 1/2 Hoagland culture medium was counted manually.

### 4.3. Determination of the Contents of Malondialdehyde, Soluble Protein, and Superoxide Anion in the Blade 

The method developed by Hodges et al. was employed to measure the levels of malondialdehyde (MDA) [43]. Frist, 1.00 g was homogenized by grinding with 2 mL of 10% Trichloroacetic acid (TCA) solution. Subsequently, an additional 8 mL of the 10% TCA solution was added to continue the grinding process. The homogenization solution was then subjected to centrifugation at 4000 rpm/min for 20 min. After centrifugation, 2.0 mL of the supernatant was collected and mixed with 2 mL of 0.6% TBA (Thiobarbituric acid) solution. The resulting mixture was boiled in a water bath for 15 min and then centrifuged again after cooling. The absorbance of the supernatant was measured at 600 nm, 450 nm, and 532 nm [43].

The method developed by Wang et al. was employed to measure the levels of soluble protein content [44]. Frist, 1.00 g of fresh blades was ground with 10 mL of 50 mmol phosphate buffer (pH 7.8) and homogenized. The resulting mixture was then centrifuged at 4000 rpm/min for 20 min. After that, 0.1 mL of the supernatant was taken and combined with 5 mL of Coomasie bright Blue G-250 reagent. After thorough mixing, the absorbance at 595 nm was measured after 2 min.

The method developed by Cui et al. was employed to measure the levels of superoxide anion [45]. The fresh blades (1.00 g) were homogenized with 10 mL of 50 mmol phosphate buffer (pH 7.8) and centrifuged at 4000 rpm/min for 20 min. Then, 0.5 mL of the supernatant was taken and combined with 0.5 mL of 50 mmol phosphate buffer (pH 7.8) and 1.0 mL of 1.0 mmol/L hydroxylamine hydrochloride. The mixture was thoroughly mixed and kept warm in a water bath at 25 °C for 1 h. Next, 1.0 mL of 17.0 mmol/L p-aminobenzenesulfonic acid and 1.0 mL of 7.0 mmol/L a-naphthylamine were added to the mixture, which was then mixed and kept warm in a water bath at 25 °C for 20 min. Finally, the absorbance at 530 nm was measured [45].

### 4.4. Chlorophyll Content Determination

For the determination of the chlorophyll content, the method described by Zhou et al. was followed [46]. Frist, from 0.20 g of fresh blades, the veins were removed and cut into thin filaments of about 0.2 cm. These thin filaments were put into 10 mL of anhydrous ethanol and acetone (1:1) mixed reagent, dark leached for 24 h, and centrifuged to remove the supernatant. Finally, the absorbance at 663 nm, 646 nm, and 470 nm was measured [46].

### 4.5. cDNA Library Construction and RNA-Seq

The drought-resistant genotype S01 and drought-sensitive genotype S26 were placed in 1/2 Hoagland culture for 48 h and 20% PEG culture for 48 h (three biological replicates for each genotype); then, the third blade was selected for RNA-seq. Total RNA was extracted using the total RNA Extraction Kit (DP419, TIANGEN, Beijing, China), the RNA concentration was measured using the Nanodrop (IMPLEN, Westlake Village, CA, USA), and RNA integrity and purity were accurately assessed using the Aglient 5400 (Agilent Technologies, Santa Clara, CA, USA). After the total RNA samples passed the test, the cDNA libraries were constructed and quality controlled by Novogene Co., Ltd. (Beijing, China). The constructed libraries were sequenced with Illumina HiSeqTM platform for transcriptome sequencing by Novogene Co., Ltd.

### 4.6. Transcriptome Assembly and Functional Annotation

The SolexaQA package filters raw reads to produce clean reads. HISAT2 constructed an index of the reference genome and mapped clean reads to the sweetpotato (Taizhong 6) genome for comparative analysis (http://public-genomes-ngs.molgen.mpg.de/SweetPotato/, accessed on 1 November 2021).

### 4.7. Differentially Expressed Genes (DEGs) Analysis

We used feature Counts software(1.5.0−p3) to perform FPKM (Fragments Per Kilobase of transcript sequence per Millions base pairs sequenced) analysis of gene expression levels. DESeq2 software (1.20.0) compared the differentially expressed genes between the combinations for statistical analysis. The cluster Profiler software (3.8.1) enabled GO (http://geneontology.org, accessed on 21 August 2023) and KEGG (https://www.genome.jp/kegg/, accessed on 21 August 2023) enrichment analysis of differentially expressed genes.

### 4.8. Weighted Gene Co-Expression Network Analysis (WGCNA)

The outlier samples were filtered by expression matrix correlation, and the gene co-expression network was analyzed by the R software WGCNA (https://horvath.genetics.ucla.edu/html/CoexpressionNetwork/Rpackages/WGCNA/, accessed on 21 August 2023) package [47]. The automatic network building function was used to obtain co-representation modules, and the correlation between modules and processing was calculated to obtain the eigenvalues of each module. Cytoscape software (3.9.1) visualized the gene co-expression regulatory networks within target modules.

### 4.9. Quantitative Real-Time PCR (qRT-PCR) Analysis of DEGs

In order to verify the reliability of RNA-seq, 9 genes in DEGs were selected for qRT-PCR. Primer Premier 5.0 was used to design the specific primers (Appendix A). *IbACTIN* was used as the internal reference gene; iTaq Universal SYBR Green Supermix (Bio-Rad, Hercules, CA, USA) was used for PCR amplification and DNA staining; and qRT-PCR was performed on an IQ5 thermal cycler (Bio-Rad, Hercules, CA, USA). The relative gene expression levels were calculated according to the 2^−ΔΔCt^ method [48].

### 4.10. Statistical Analysis

Microsoft Excel 2021 was used for data statistics and calculation of the biological repeatable mean and standard errors; SPSS software (IBM SPSS Statistics version 26, Chicago, Illinois) was used for the analysis of variance (ANOVA) and correlation analysis; GraphPad Prism 9 was used for chart drawing; and TBtools [49] was used for heatmap plotting. A *t*-test was used for significance analysis, with *p*-value ≤ 0.05 as the statistically significant level. The data were presented as the mean ± standard error (SE) from three independent biological samples.

## 5. Conclusions

In this study, for 27 sweetpotato genotypes, the RWC, VTFWR, and the R of the vine tip were determined, and several strong drought-resistant types were identified, including S01 (XS161819), S04 (Mianshu-6), S02 (XN1729-11), S07 (Jinshu-3), S05 (21-F-3), S10 (21-P-29), and S08 (G20-9). Genotypes S01 (XS161819) and S26 (18-12-3) were selected to verify their significant differences to drought resistance at physiological and RNA transcription levels. The hormone signal transduction pathway was one of the most critical pathways affecting drought resistance and midnight blue modules with a high positive correlation with the drought-resistance physiological characteristics of sweetpotato were determined. The top 20 Hub genes with the largest betweenness centrality were selected, including *g47370* (*AFP2*), *g14296* (*CDKF*), and *g60091* (*SPBC2A9*). The study screened out excellent lines or germplasms for the breeding of drought-resistant sweetpotato and provided a theoretical basis for the potential excavation of drought-resistant genes in sweetpotato.

## Figures and Tables

**Figure 1 ijms-24-14398-f001:**
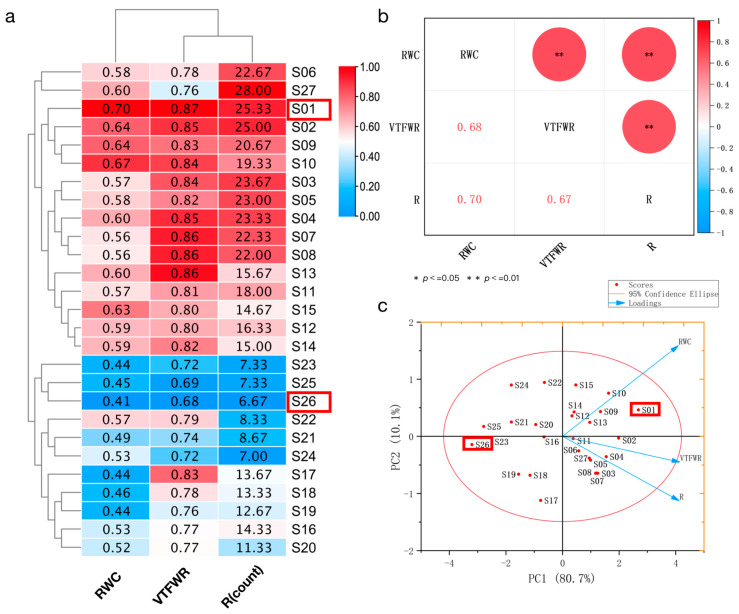
RWC, VTFWR, and R heat map, correlation analysis, and principal component analysis for 27 genotypes. (**a**) RWC, VTFWR, and R heat map of 27 sweetpotato genotypes. (**b**) Correlation analysis between RWC, VTFWR, and R. (**c**) Principal component analysis of 27 Sweetpotato genotypes. The values in Figure (**a**) are the RWC, VTFWR, and the R of vine tip from 27 sweetpotato genotypes; this refers to the heatmap of the maximum value of each column. The red box outlines the drought-resistant genotype S01 and the drought-sensitive genotype S26. In (**b**), * represents *p* < 0.05, and ** represents *p* < 0.01. In (**c**), the red box outlines the drought-resistant genotype S01 and the drought-sensitive genotype S26.

**Figure 2 ijms-24-14398-f002:**
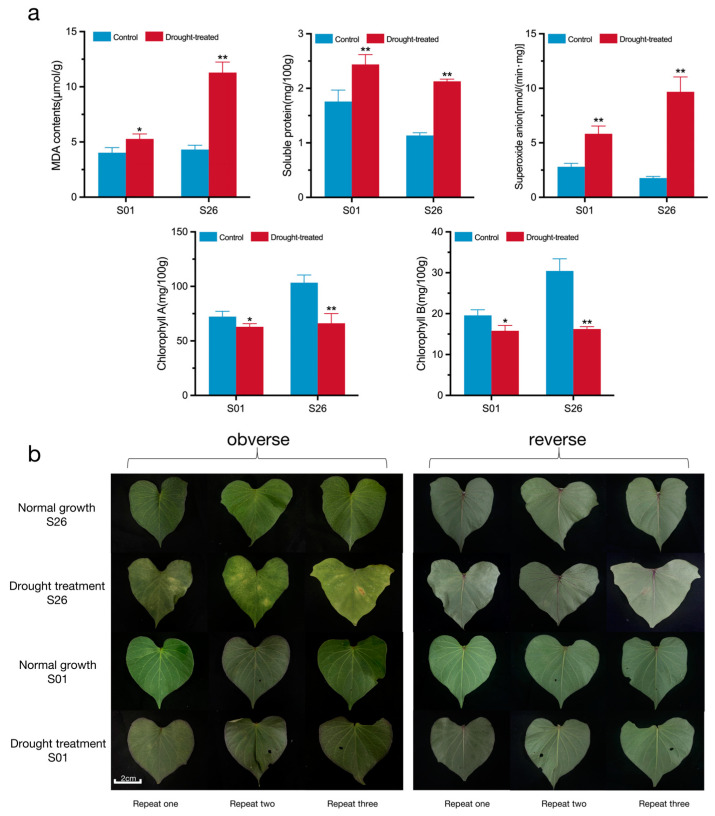
Physiological indicators and blade phenotypic changes in S01 and S26. (**a**) Determination of physiological indicators for drought resistance in S01 and S26. (**b**) Phenotypic changes in S01 and S26 blades under simulated drought conditions. In (**a**), * and ** indicate that the differences in the physiological index between the control group and experimental group are significant at *p* < 0.05 and *p* < 0.01, respectively.

**Figure 3 ijms-24-14398-f003:**
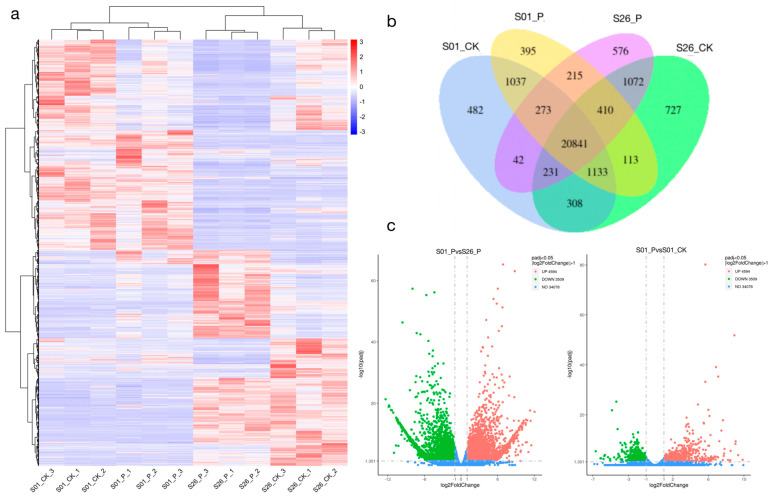
Heat map, Venn map, and volcano map of DEGs. (**a**) Cluster analysis of DEGs. (**b**) Number of DEGs among different groups. (**c**) Volcanic maps of S01_P and S26_P, and S01_P and S01_CK. Figure (**a**) S01_P, S01 experimental group; S01_CK, S01 control group; S26_P, S26 experimental group; and S26_CK, S26 control group. Figure (**c**) NO, all the DEGs; UP, upregulated DEGs; and DOWN, downregulated DEGs.

**Figure 4 ijms-24-14398-f004:**
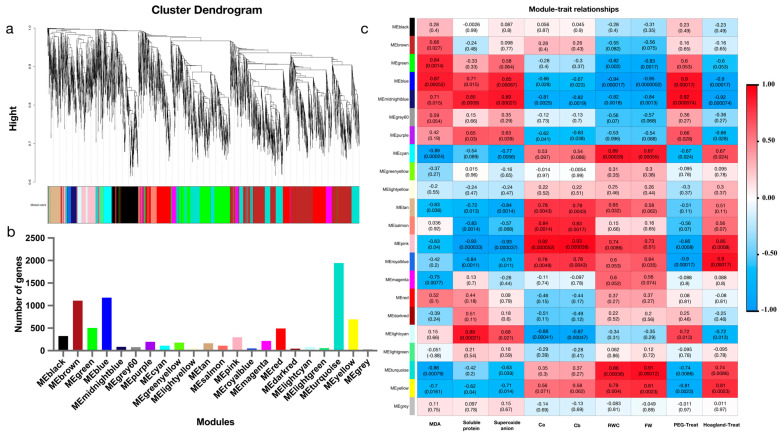
Key gene modules and Hub genes for drought tolerance screened by WGCNA. (**a**) Cluster dendrogram of DEGs based on WGCNA analysis. (**b**) Number of genes in each module. (**c**) Correlation analysis between gene modules and traits.

**Figure 5 ijms-24-14398-f005:**
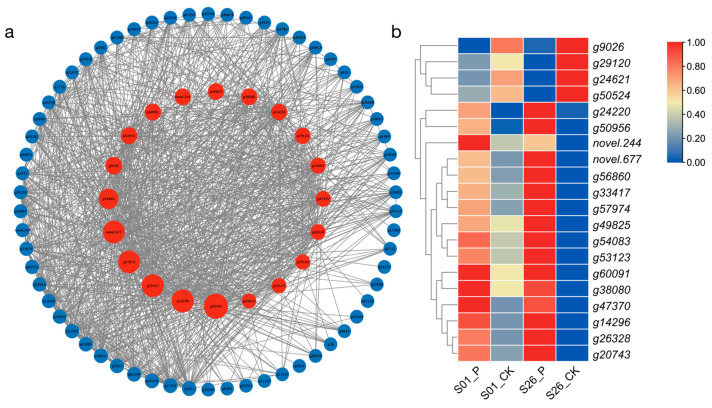
Selected WGCNA-selected drought-tolerant Hub genes from midnight blue modules. (**a**) Hub genes discovered in midnight blue. (**b**) The expression level of Hub genes in RNA-seq. Figure (**a**) the red circle represents the Hub genes; the size of the circle represents the size of the betweenness centrality.

**Figure 6 ijms-24-14398-f006:**
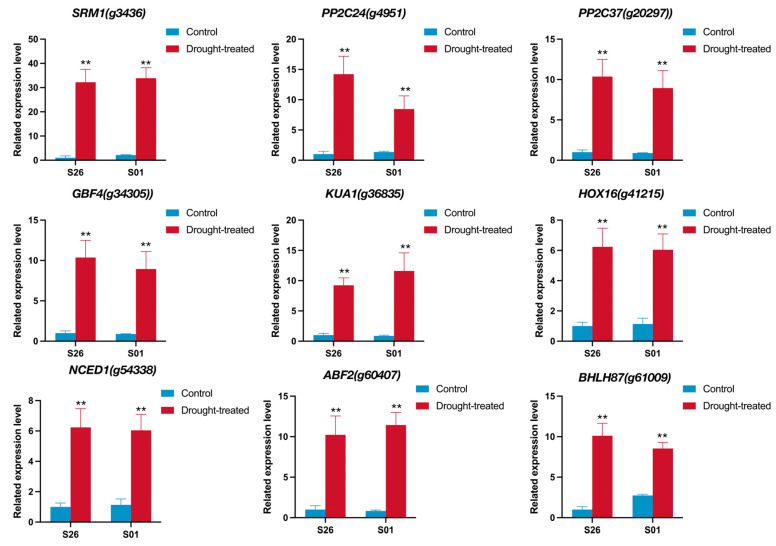
Verifying RNA-seq data through qRT-PCR. ** indicates that the differences in the gene-related expression between the control group and the experimental group are significant at *p* < 0.01.

## Data Availability

The datasets generated during and/or analyzed during the current study are available from the National Center for Biotechnology Information repository, (https://www.ncbi.nlm.nih.gov/bioproject/PRJNA999504, accessed on 28 July 2023).

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
