# Peer review of "Transcriptome-Based WGCNA Analysis Reveals the Mechanism of Drought Resistance Differences in Sweetpotato (Ipomoea batatas (L.) Lam.)"

_ijms, 2023, doi:10.3390/ijms241814398_

Round 1

Reviewer 1 Report

The manuscript entitled ‘Comparative Transcriptome Gene Co-expression Network Analysis Confirmed the Sweet potato (Ipomoea batatas (L.) Lam.) Hub Genes Related to Drought Tolerance’ describes the screening and selection for drought resistance germplasm and insights on the genetic basis of drought tolerance mechanism in Sweet potato, an economically important crop. A drought resistant and susceptible variety was identified by PEG based assay based on phenotype and physiological measurements. The differential gene expression was analyzed based on leaf RNA sequencing and co-expression analysis revealed gene expression modules leading to the identification of 20 Hub genes that are potentially important for drought response.

Suggested revisions:

1.    The manuscript needs to be thoroughly checked for English language, spelling and grammar. (Lines 81,86,91,218, 243, 246, 249 as examples)

2.    The selection of resistant Vs susceptible line is based on the 48hr 20% PEG treatment. Has the drought tolerant phenotype in S01 been verified by imposing drought stress at the whole plant level? Although the main emphasis of this article is on understanding the molecular basis of drought response, it is important to verify this for future applicability of this assay for high throughput screening applications.

3.    The abstract section quotes ‘twenty two differential expression modules two of which showed a strong positive correlation with drought resistance characteristics’, however there is mention of only 1 (midnight blue) in the text. A light yellow module is mentioned in the conclusion but not in results. Are all the hub genes identified from the midnight blue module? The authors need to clarify this and make necessary changes wherever applicable.

 The manuscript needs to be thoroughly checked for English language, spelling and grammar. (Lines 81,86,91,218, 243, 246, 249 as examples)

Author Response

Many thanks indeed for your kind and professional evaluation of our manuscript (ijms-2599362) entitled “Transcriptome-Based WGCNA Analysis Reveals the Mechanism of Drought Resistance Differences in Sweetpotato (Ipomoea batatas (L.) Lam.)”! We have carefully read and revised this manuscript accordingly, and the fixed parts are highlighted in yellow. Finally, the revised manuscript and related materials were uploaded to the International Journal of Molecular Sciences. I want to address your evaluation and comments point by point as below.

REVIEWER’S COMMENT 1: The manuscript needs to be thoroughly checked for English language, spelling and grammar. (Lines 81,86,91,218, 243, 246, 249 as examples)

Author’s Response: Thank you for your reminder; we have checked the English grammar and spelling of the entire text.

REVIEWER’S COMMENT 2: The selection of resistant Vs susceptible line is based on the 48hr 20% PEG treatment. Has the drought tolerant phenotype in S01 been verified by imposing drought stress at the whole plant level? Although the main emphasis of this article is on understanding the molecular basis of drought response, it is important to verify this for future applicability of this assay for high throughput screening applications.

Author’s Response: Thank you for your reminder; we have added a phenotypic map of drought treatment applied at the entire plant level in the attached figure section.

REVIEWER’S COMMENT 3: The abstract section quotes ‘twenty two differential expression modules two of which showed a strong positive correlation with drought resistance characteristics’, however there is mention of only 1 (midnight blue) in the text. A light yellow module is mentioned in the conclusion but not in results. Are all the hub genes identified from the midnight blue module? The authors need to clarify this and make necessary changes wherever applicable.

Author’s Response: Many thanks for your kind comments! We have modified the abstract and conclusion sections, and all hub genes have been identified from the midlightblue module.

The responses above are long, and we appreciate your kind and professional comments on our work, which are very helpful in improving our manuscript. If there is any problem with this manuscript, please get in touch with me. We are looking forward to your response.

Many thanks indeed, and best regards!

Reviewer 2 Report

In this manuscript, Zong et al. initially screened a panel of 27 sweet potato lines and then, based on differential responses to drought-mimicking PEG treatments, selected two lines (S01 and S26) for more physiological comparisons and in-depth transcriptomics analysis. In the physiological responses to drought, they identified that line named S26 showed higher sensitivity to water limitation with faster senescence and lowered chlorophyll content compared to line S01. Further, they then performed RNAseq analysis using RNA collected from leaf blades of S01 and S26 from the experimental (treated with 20% PEG6000 for 48h), compared with the control group. In the WGCNA analysis, they identified 22 expression modules that correlated with the differential responses to PEG, and they verified some of the DEG of selected genes in one module by qPCR analysis, supporting RNAseq results. In the discussion part, the authors highlight some involvement of genes related to hormonal regulation, including differentially expressed genes in auxin and abscisic acid pathways and more specifically, elevated Aux/IAA gene expression in the more drought-tolerant S01 line.

General and significant comments:

Overall, this study uses a rationale workflow while screening a panel and focusing on two extremely different lines for their responses to drought for more in-depth transcriptome analysis. To make this choice they used many physiological tests; however the terms RWC, VTFWR, R, MDA, soluble protein content, superoxide anion, and chlorophyll content is mentioned in any way, shape or form in the introduction and are hardly referenced in the results. They should include a reference to these before they present this in the results. Though cursory mention of them exists in the introduction, it needs to have the effect that makes it clear that these will be used in drought tolerance quantification.

Similarly:

278-296: first time ABA and salicylic acid is mentioned since the introduction. Where was this shown in the results? Also, the authors are introducing thoughts and ideas that they did not touch on in the introduction.

251-263: Many of these explanations are coming in very late.

It helps the reader should this explanation be present before the results; otherwise, the results are difficult to understand.

The manuscript is very poorly written. The English could be clearer to understand . In some cases, it is unclear why they choose to include sentences, and in some sentences, this has simple repetition of words as it was not proofread before submission (see some examples below). Also, it is recommended not to include lengthy sentences such as 267-272 and consider breaking them to two or more shorter ones. I recommend sending it to language editing to improve its clarity.

In some places, the authors jump directly into a description of the results with no or little explanation as to the experiment being performed, and why it was conducted. A brief overview of the experiment, and even its purpose, could help the reader better understand their point. Furthermore, acronyms are given without any explanation. For example, row 109.

Part 2.3 (155-165) describes a process (generation of RNAseq) rather than showing results. Consider a better balance between Methods and Results.

The authors performed WGCNA and identified 22 clusters that they correlated with the different drought responses. They choose to name the clusters as color-coding models; therefore, modules get strange and hard-to-follow names, e.g., midlightblue. Why not name them by numbers and call them, for example, module 1 to module 22?

Minor comments:

Figure 1: no units are given for figure a.

113: there is a mention of "genotype type". This term stands unexplained. Also, check grammar, i.e.they had no difference i

109-122: In the text there is reference to results with no units.

132-147: why absolute and percentage?

188-189: what of S01_CK?

Row 192: was it a 2 or 4 way analysis? Ambiguous. Not clear what is being said here. What is the point the writer is trying to get across?

194-195: "“Plant hormone signal transduction” and “amino acid biosynthesis”, they played an important role in drought resistance of  sweetpotato."- several groups were mentioned. Why are these identified?

100: what is FKPM?

201: WGCNA?

212: what is parallel correlation- do they mean positive one?

216: Median centrality?

217: " greatest betweeness centrality"??? betweeness=not a word

222-223: Why does their higher expression in the controls matter?

232-238: is this strictly necessary?

246: The word drought appears 3 times in a row

298: indicated 23 modules, while previously 22. Check. here 23

Conclusions:

260-264: needs to be clarified what is being said here.

To conclude, the question of what genes and processes are involved in sweet potato drought resistance are indirectly answered but portrayed on several physiological levels, and with a good WGCNA analysis. A focused concluding paragraph focusing on what genes and processes are involved would drive the point home and clarify it.

See within comments to author

Author Response

Many thanks indeed for your kind and professional evaluation of our manuscript (ijms-2599362) entitled “Transcriptome-Based WGCNA Analysis Reveals the Mechanism of Drought Resistance Differences in Sweetpotato (Ipomoea batatas (L.) Lam.)”! We have carefully read and revised this manuscript accordingly, and the fixed parts are highlighted in yellow. Finally, the revised manuscript and related materials were uploaded to the International Journal of Molecular Sciences. I want to address your evaluation and comments point by point as below.

GENERAL AND SIGNIFICANT COMMENTS:

REVIEWER’S COMMENT 1: Overall, this study uses a rationale workflow while screening a panel and focusing on two extremely different lines for their responses to drought for more in-depth transcriptome analysis. To make this choice they used many physiological tests; however, the terms RWC, VTFWR, R, MDA, soluble protein content, superoxide anion, and chlorophyll content is mentioned in any way, shape or form in the introduction and are hardly referenced in the results. They should include a reference to these before they present this in the results. Though cursory mention of them exists in the introduction, it needs to have the effect that makes it clear that these will be used in drought tolerance quantification.

251-263: Many of these explanations are coming in very late.

It helps the reader should this explanation be present before the results; otherwise, the results are difficult to understand.

Author’s Response: Thanks for your helpful comments. We have provided a detailed introduction to various indicators in the introduction. Include partial explanations from lines 251 to 263 in the introduction to help readers better understand the article.

REVIEWER’S COMMENT 2: 278-296: first time ABA and salicylic acid is mentioned since the introduction. Where was this shown in the results? Also, the authors are introducing thoughts and ideas that they did not touch on in the introduction.

Author’s Response: We are very grateful for your kind comments! We have added a relevant description in the 2.4 Results section and a more detailed description of WGCNA in the Introduction.

REVIEWER’S COMMENT 3:The manuscript is very poorly written. The English could be clearer to understand. In some cases, it is unclear why they choose to include sentences, and in some sentences, this has simple repetition of words as it was not proofread before submission (see some examples below). Also, it is recommended not to include lengthy sentences such as 267-272 and consider breaking them to two or more shorter ones. I recommend sending it to language editing to improve its clarity.

Author’s Response: Thank you for your reminder; we have checked the grammar of the entire text and sent it to the MDPI language editor to improve clarity.

REVIEWER’S COMMENT 4: In some places, the authors jump directly into a description of the results with no or little explanation as to the experiment being performed, and why it was conducted. A brief overview of the experiment, and even its purpose, could help the reader better understand their point. Furthermore, acronyms are given without any explanation. For example, row 109.

Author’s Response: Thanks for your kind suggestion! We added a description of the purpose of the experiment for Results 2.1 and 2.4 section. We checked for the absence of explanations for the complete text acronyms and added illustrations.

REVIEWER’S COMMENT 5: Part 2.3 (155-165) describes a process (generation of RNAseq) rather than showing results. Consider a better balance between Methods and Results.

Author’s Response: Thank you for your careful review of our manuscript. We have streamlined section 2.3 of the results and removed descriptions of biological duplication.

REVIEWER’S COMMENT 6: The authors performed WGCNA and identified 22 clusters that they correlated with the different drought responses. They choose to name the clusters as color-coding models; therefore, modules get strange and hard-to-follow names, e.g., midlightblue. Why not name them by numbers and call them, for example, module 1 to module 22?

Author’s Response: Thank you for your reminder because the earliest authoritative literature was the R package using color coding modules. The Cluster Dendrogram diagram cannot express the meaning well if a digital encoding module is used.

MINOR COMMENTS:

REVIEWER’S COMMENT 1: Figure 1: no units are given for figure a.

Author’s Response: Thank you for your kind comments. The RWC and VTFWR in Figure 1a are ratios without units, and we have added teams to the R.

REVIEWER’S COMMENT 2: 113: there is a mention of "genotype type". This term stands unexplained. Also, check grammar, i.e.they had no difference i

Author’s Response: We apologize for the language-related errors in the original manuscript.  We removed 'type' and conducted a detailed check on the sentence.

REVIEWER’S COMMENT 3: 109-122: In the text there is reference to results with no units.

Author’s Response: Many thanks for your kind comments! Most of them are ratios without units.

REVIEWER’S COMMENT 4: 132-147: why absolute and percentage?

Author’s Response: Thank you for your reminder; we have standardized the format and changed it to absolute values.

REVIEWER’S COMMENT 5: 188-189: what of S01_CK?

Author’s Response: Thanks for your kind suggestion! S01_CK means S01 genotype control group, and we explain the meaning of S01_CK, S01_P, S26_CK, S26_P in detail in the note to Figure 3.

REVIEWER’S COMMENT 6: Row 192: was it a 2 or 4 way analysis? Ambiguous. Not clear what is being said here. What is the point the writer is trying to get across?

Author’s Response: Many thanks for your kind comments! We have removed the section 'gene weighted network co expression analysis was performed on all transcriptome samples to screen out key gene modules of sweetpotato drop tolerance. First,' to make the meaning of the article more concise.

REVIEWER’S COMMENT 7: 194-195: "Plant hormone signal transduction” and “amino acid biosynthesis”, they played an important role in drought resistance of sweetpotato."- several groups were mentioned. Why are these identified?

Author’s Response: Many thanks indeed for your kind suggestion! Because KEGG enrichment analysis found that the genes enriched in these two pathways are the most.

REVIEWER’S COMMENT 8: 100: what is FKPM?

Author’s Response: Thank you for your careful review of our manuscript. I'm very sorry for the spelling error caused by my negligence. This should be FPKM (Fragments Per Kilobase of script sequence per Millions base pairs sequence).

REVIEWER’S COMMENT 9: 201: WGCNA?

Author’s Response: Thanks for your helpful comments. The full name of WGCNA is Weighted correlation network analysis, which was mentioned in the introduction, so it is abbreviated here.

REVIEWER’S COMMENT 10: 212: what is parallel correlation- do they mean positive one?

Author’s Response: Thank you for your reminder; we are very sorry for the inappropriate use of words here. We have changed 'parallel correlation' to 'positive correlation.’

REVIEWER’S COMMENT 11: 216: Median centrality?

Author’s Response: Many thanks for your kind comments! We are very sorry for the inappropriate use of words here. We have modified 'Median centrality' to 'betweenness centrality,’ where 'betweenness centrality' is a proper noun.

REVIEWER’S COMMENT 12: 217: "greatest betweeness centrality"??? betweeness=not a word

Author’s Response: Thanks for your helpful comments. We have made modifications to the entire text by changing 'betweeness' to 'betweenness'.

REVIEWER’S COMMENT 13: 222-223: Why does their higher expression in the controls matter?

Author’s Response: We are very grateful for your kind comments! High expression in the control indicates that it can respond more quickly to drought treatment, which may be one of the reasons for the difference in drought resistance between the two varieties.

REVIEWER’S COMMENT 14: 232-238: is this strictly necessary?

Author’s Response: Thank you for your careful review of our manuscript. Mainly introduced the current situation of sweetpotato in the global arid environment and discussed the main ways to reduce the impact of drought on sweetpotato yield.

REVIEWER’S COMMENT 15: 246: The word drought appears 3 times in a row

Author’s Response: We apologize for the language-related errors in the original manuscript. We have condensed the sentence.

REVIEWER’S COMMENT 16: 298: indicated 23 modules, while previously 22. Check. here 23

Author’s Response: I'm sorry, we have changed the "23" in the discussion section to "22".

REVIEWER’S COMMENT 17: Conclusions: 260-264: needs to be clarified what is being said here.

Author’s Response: Thanks for your kind suggestion! We checked the conclusion and changed "blue and lightyellow modules" to "midnightblue module".

The responses above are long, and we appreciate your kind and professional comments on our work, which are very helpful in improving our manuscript. If there is any problem with this manuscript, please get in touch with me. We are looking forward to your response.

Many thanks indeed, and best regards!

Reviewer 3 Report

Comments

In general, the submitted manuscript is set out to assess the gene network invloved in drought stress tolerance and/or resistance in different sweetpotato cultivrs. I trust that the paper has the publication potential, however it should be improved in various aspects, as mentioned in the following comments:

Title

In my opinion, title is too generic and should be modified as it does not contain the enough information of the study.

Abstract

Abstract is overall fine. However, the results in abstract are not clear. It would better if you can add some exact numbers of some of the key findings of the currnet study.

Introduction

Overall, the Introduction is fine, however few minor changes are required:

1-The objectives of the study are not clearly written. Please revise the sentences.  

2-There are several poorly structured and unclear sentences. Please improve the sentence structure and language of your paper.

Results:

1-In results section, subheading “2.1: Selection of drought resistant sweetpotato genotypees” need some modifications. Please add the exact number (% increase or decrease) of the key findings.

2-Subheading “2.2” also need some changes. For example line 140 and 143 authors mentioned “672.74%, 226.32% increase” isn’t it too much? Similarly, in the same paragraph authors mentioned “3.76 times”. Please be comsistant and use only one style. However, 672% is not recommended, use the fold change term for these type of values.

3-Figure 4 (c): figure quality is poor. It is recommede to add/make a new high quality figure of correlation analysis.

Discussion:

1-Discussion section need thorough revision. Authors focused on the current results while writing this section. Authors discussed their own results and added some references at the end of the sentence. I would suggest revise this section and compare your results with the Previous findings either the current findings corrobrate with the previous studies or not.

2- Some grammatical errors were also noticed. Please improve the sentence structure and language of your paper.

Materials and Method:

1-Statistical analysis (section 4.10), is not clear. Please add details about statistical design, least significant different test or level of probablity, replications, and treatments.

Conclusion:

1- I have found some poorly written sentences (for example: vary last sentence line 427-429). Please recheck the sentence structure and grammar of your manuscript before resubmitting it for the possible publication.

Minor editing of English language required

Author Response

Many thanks indeed for your kind and professional evaluation of our manuscript (ijms-2599362) entitled “Transcriptome-Based WGCNA Analysis Reveals the Mechanism of Drought Resistance Differences in Sweetpotato (Ipomoea batatas (L.) Lam.)”! We have carefully read and revised this manuscript accordingly, and the fixed parts are highlighted in yellow. Finally, the revised manuscript and related materials were uploaded to the International Journal of Molecular Sciences. I want to address your evaluation and comments point by point as below.

REVIEWER’S COMMENT 1: In my opinion, title is too generic and should be modified as it does not contain the enough information of the study.

Author’s Response: Thank you for your reminder; we have revised the original title 'Comparative Transcriptome Gene Co-expression Network Analysis Confirmed the Sweetpotato (Ipomoea batatas (L.)  Lam.)  Hub Genes Related to Drought Tolerance' to 'Transcriptome Based WGCNA Analysis Reveals the Mechanism of Drought Resistance Differences in Sweetpotato (Ipomoea batatas (L.)  Lam.)'.

REVIEWER’S COMMENT 2: Abstract is overall fine. However, the results in abstract are not clear. It would better if you can add some exact numbers of some of the key findings of the current study.

Author’s Response: Thanks for your helpful comments. We have added the exact number of key findings from recent research and modified it to 'The results showed that the relevant water content (RWC) and vine tip fresh weight reduction (VTFWR) in XS161819 were 1.17 and 1.14 times higher than the recognized drought-resistant variation Chaoshu 1.'.

REVIEWER’S COMMENT 3: 1-The objectives of the study are not clearly written. Please revise the sentences. 

Author’s Response: We are very grateful for your kind comments! We added a detailed purpose at the end of the introduction; the differences in the resistance mechanisms to drought in two sweetpotato genotypes were further explored by an RNA-Seq analysis Using WGCNA technology to associate physiological indicators such as the RWC, VTFWR, and MDA with gene expression patterns, we extract hub genes directly related to drought defense, and constructed a co-expression network.

REVIEWER’S COMMENT 4: 2-There are several poorly structured and unclear sentences. Please improve the sentence structure and language of your paper.

Author’s Response: Many thanks indeed for your kind suggestion! We have made modifications to the sentence structure and grammar of the entire text and submitted it to the MDPI editor for polishing.

REVIEWER’S COMMENT 5: 1-In results section, subheading “2.1: Selection of drought resistant sweetpotato genotypees” need some modifications. Please add the exact number (% increase or decrease) of the key findings.

Author’s Response: "The RWC, VTFWR, and R of S01 are 1.71 times, 1.28 times, and 3.79 times that of S26" to the 2,1 result section.

REVIEWER’S COMMENT 6: 2-Subheading “2.2” also need some changes. For example line 140 and 143 authors mentioned “672.74%, 226.32% increase” isn’t it too much? Similarly, in the same paragraph authors mentioned “3.76 times”. Please be consistentand use only one style. However, 672% is not recommended, use the fold change term for these types of values.

Author’s Response: Thanks for your helpful comments. We have modified all types to resemble the "3.76 times" mode.

REVIEWER’S COMMENT 7: 3-Figure 4 (c): figure quality is poor. It is recommended to add/make a new high quality figure of correlation analysis.

Author’s Response: Thank you for your reminder; we have remade a more transparent and more aesthetically pleasing image.

REVIEWER’S COMMENT 8: 1-Discussion section need thorough revision. Authors focused on the current results while writing this section. Authors discussed their own results and added some references at the end of the sentence. I would suggest revise this section and compare your results with the Previous findings either the current findings corrobrate with the previous studies or not.

Author’s Response: We are very grateful for your kind comments! We have removed many descriptions of specific indicators, added comparisons with other studies, and discussed more novel viewpoints in this article, such as discussions on “Carbon metabolism,” “Biosynthesis of amino acids,” and “Starch and sucrose metabolism.”

REVIEWER’S COMMENT 9: 2- Some grammatical errors were also noticed. Please improve the sentence structure and language of your paper.

Author’s Response: Many thanks indeed for your kind suggestion! We have modified the sentence structure and grammar of the entire text and submitted it to the MDPI editor for polishing.

REVIEWER’S COMMENT 10: 1-Statistical analysis (section 4.10), is not clear. Please add details about statistical design, least significant different test or level of probablity, replications, and treatments.

Author’s Response: Thank you for your careful review of our manuscript. We have modified sections 4.4 and 4.10 of the material method by adding content such as statistical design and the least significant difference test.

The responses above are long, and we appreciate your kind and professional comments on our work, which are very helpful in improving our manuscript. If there is any problem with this manuscript, please get in touch with me. We are looking forward to your response.

Many thanks indeed, and best regards!

Round 2

Reviewer 1 Report

The authors have addressed the comments and revised the manuscript.